# A Recent and Systematic Review on Water Extraction from the Atmosphere for Arid Zones

**Yinyin Wang** [1,2], **Suad Hassan Danook** [3], **Hussein A.Z. AL-bonsrulah** [4,*], **Dhinakaran Veeman** [5] **and Fuzhang Wang** [6]

1    Jiangsu Yangtze River Economic Belt Research Institute, Nantong University, Nantong 226002, China; wyyntu11@163.com
2    Economics and Management School, Nantong University, Nantong 226019, China
3    Kirkuk Technical College, Northern Technical University, Nineveh 41002, Iraq; suaddanook@ntu.edu.iq
4    Department of Mechanical Engineering, Faculty of Engineering, Kufa University, Najaf 54002, Iraq
5    Centre for Computational Mechanics, Chennai Institute of Technology, Chennai 600069, India; dhinakaranv@citchennai.net
6    Nanchang Institute of Technology, Nanchang 330044, China; wangfuzhang1984@163.com
*    Correspondence: huseenabd541@gmail.com

**Abstract:** Water is essential for food security, industrial output, ecological sustainability, and a country's socioeconomic progress. Water scarcity and environmental concerns have increased globally in recent years as a result of the ever-increasing population, rapid industrialization and urbanization, and poor water resource management. Even though there are sufficient water resources, their uneven circulation leads to shortages and the requirement for portable fresh water. More than two billion people live in water-stressed areas. Hence, the present study covers all of the research based on water extraction from atmospheric air, including theoretical and practical (different experimental methods) research. A comparison between different results is made. The calculated efficiency of the systems used to extract water from atmospheric air by simulating the governing equations is discussed. The effects of different limitations, which affect and enhance the collectors' efficiency, are studied. This research article will be very useful to society and will support further research on the extraction of water in arid zones.

**Keywords:** absorption; solar energy; dew collection; desiccant; regeneration

## 1. Introduction

Freshwater production is one of the world's most pressing issues today. Despite the fact that desalination can provide a climate-independent source of clean water, it is a time-consuming and energy-intensive process [1,2]. Even though there are sufficient water resources, their uneven circulation leads to shortages and the requirement for portable fresh water. More than two billion people live in water-stressed areas. Large-scale desalination systems, which are currently largely powered by nonrenewable energy sources, have gradually improved freshwater supplies in several coastal regions [3]. In addition, water is used inefficiently in many fields, such as industries, agriculture, and many others. Currently, many countries in the world are facing water crises [4]. As of now, the percentage of fresh water available on the Earth's surface is precisely 3.0%, and the remaining 97% is saline water. A considerable fraction of this fresh water persists in the form of ice (68.7%). This is mainly the case in regions of the Arctic and Antarctica. The primary water use is only 0.26% (rivers and freshwater lakes) of the entire global water reserves (90,000 km$^3$) for human consumption [5,6].

The air may be used as a sustainable water supply because it contains over 14,000 km$^3$ of water vapor [7]. There are two methods for extracting water from ambient air. The first approach entails lowering the temperature of humid ambient air to below the dew

point. The second method entails sucking water vapor from moist ambient air with a solid adsorbent and a liquid absorbent and then retrieving the water by heating the absorbent and liquefying the evaporated water [8]. Dew water appears to be an easy way to supplement drinking water supplies in a few parts of the world. Small animals and plants are the primary consumers of dew water, as it is necessary for maintaining their activities in a semiarid or dry environment. Engineering informs the decision on which methods to use. Economic considerations, such as energy, operational and capital costs, and climatic circumstances, also play roles. On the other hand, unique tools are different from drinking water since they are made up of small components that are sufficient for the daily needs of one individual. Large office buildings can provide portable fresh water to the surrounding community [9]. The literature review reveals that little work has been carried out to summarize the extraction of water from atmospheric air. Hence, an initiative is taken through this research work to summarize all of the possible technical methods for water extraction. This research work will help researchers and industry to move forward in water extraction from the atmosphere for arid zones.

## 2. Water Extraction Methods

Water vapor is a key component of the atmosphere in atmospheric sciences, meteorology, hydrology, and climate studies. Water vapor has an impact on the Earth's atmosphere's energy budget, which causes and maintains atmospheric motions. Solar radiation absorption, nonradiative transport from the Earth's surface (convection and condensation of water), and thermal radiation absorption all provide energy to the atmosphere [10]. In most parts of the world, 1 km$^2$ of atmospheric air contains 10,000–30,000 m$^3$ of pure water. The patented extraction of water from air (EWA) technique was designed for large-scale water supply, up to 1000 m$^3$ per day, and is based on the extraction of air humidity into a water stream. The EWA method makes use of air humidity, similar to desalination, which uses an endless free source of salty water. In regions where neither salty water nor infrastructure is available, EWA technology could be a viable alternative for water supply. EWA technology extracts air humidity in three stages: humidity absorption on a solid desiccant, water vaporization at moderate temperatures (65–85 °C), and condensation with a passive condenser connected to a heat pump. EWA technology may offer a reasonable option for water delivery in dry locations, such as the South Mediterranean, as well as countries with contaminated water, such as tropical countries, and those far from seashores where long-pipe systems are not available [11].

### 2.1. Air-to-Water Generators (AWGs)

Water extraction from air via reverse-cycle systems is becoming increasingly popular, and a variety of air-to-water generators (AWGs) are now available, all of which claim to be the most efficient [12]. The atmospheric water generator (AWG) is a possible solution for water scarcity since it converts water vapor to liquid water [13]. Figure 1 shows the experimental setup of an atmospheric water generator.

The method of water extraction from air is the same for all techniques, whether reverse cycle or not, and it involves driving the condensation of the air vapor content. AWGs with a compression reverse cycle (Figure 2), in particular, force condensation by cooling air to below the dew point. The fundamental benefit of using atmospheric water as a source of drinking water is that no water-transport infrastructure is required; harvesting equipment can be installed practically anywhere (away from the coastline). Dew condensers are divided into two categories: passive and active.

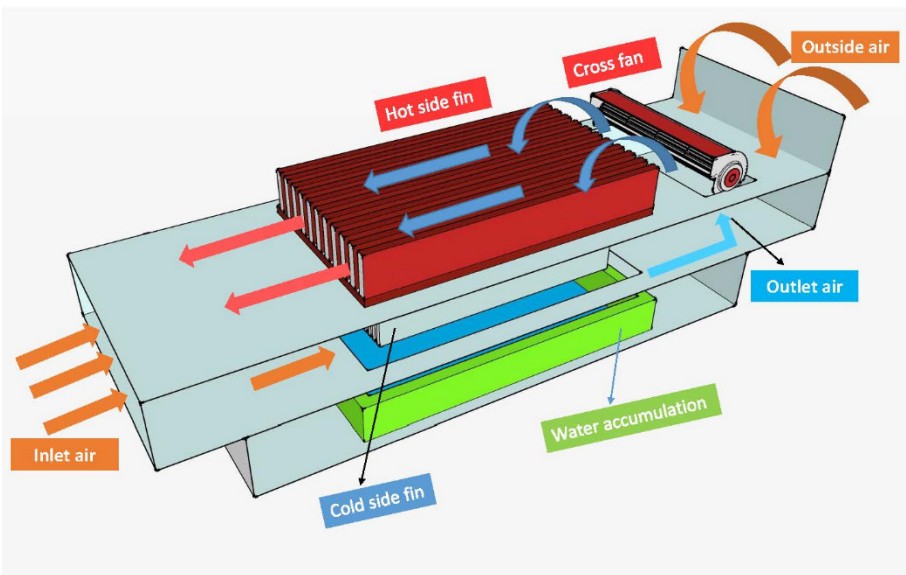

**Figure 1.** Atmospheric water generator [13] (under the terms of the Creative Commons Attribution License CC BY 4.0 license, https://creativecommons.org/licenses/by/4.0/ (accessed on 25 October 2021)).

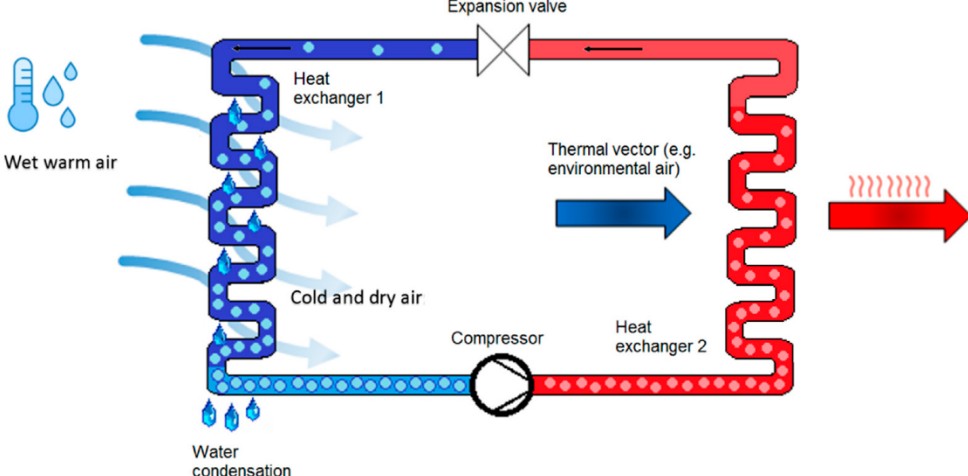

**Figure 2.** Reverse thermodynamic cycle [12] (under the terms of the Creative Commons Attribution License CC BY 4.0 license, https://creativecommons.org/licenses/by/4.0/ (accessed on 25 October 2021)).

Atmospheric water generators are another name for active condensers (AWGs). With an average airflow of 400 m$^3$/h and a compressor output of 1000 W, the unique AWG in the figure generates water at a minimum temperature of 10 °C and cools the ambient air to 8 °C below the dew point. It includes two 2 L water tanks that are completely distinct from one another, and schedules can be applied to create water for each container at a specific time. The device is 35 cm in height, 25 cm in length, and 14 cm in width [12]. Condensation is the primary method by which the AWG converts water vapor to liquid water. It cools moist air to below dew point temperatures, inducing a phase transition from vapor to liquid water across the cooling surfaces, which is subsequently collected [14,15]. The vapor-compression refrigeration cycle is used in condensation-based AWGs. The ability of the AWG to harvest water from relatively dry air and low temperatures is its most promising feature. Although relative humidity is important to the AWG's performance, it is less affected by abiotic variables such as sky emissivity, wind speed, and topographic

position than passive condensers. As a result, it may be able to operate in a wider range of weather conditions [14].

Khalil et al. conducted an inquiry to look into the refrigeration and dehumidification air-conditioning procedure as a viable method for producing fresh water and the viability of employing it in areas with high temperatures and relative humidity. Several elements, including moist air characteristics, air velocity, the surface area of the cooling coil, and heat exchange arrangement, influence the amount of condensate produced [16].

Habeebullah et al. studied the restraints of water production from the water vapor that exists in atmospheric air when cooling hot, moist air over desiccator coils of a refrigerator and then directing it to an open area. This approach can provide a limited quantity of fresh water at no cost since the water is a consequence of the air-conditioning process. A sophisticated model was used to calculate the water output due to desiccation on finned desiccator coils. Experiments were performed, and the results showed that the water productivity was reduced with an increase in the air velocity due to inadequate evaporator capacity, and it decreased at insufficient airspeed because the coolant deprivation obstructed the water extraction due to the formation of frost on the tube surface [17].

Energy was used to examine the performance and economic feasibility of coupling solar liquid desiccant dehumidification with a standard vapor-compression air-conditioning system for the meteorological scenario in Hong Kong. The capacity of the steam compression system of the solar-reinforced air-conditioning system can be reduced from the typical air-conditioning system's original performance to 19–28 kW. The performance of a standard air-conditioning system was compared to that of a solar desiccant dehumidification system. Due to higher COP resulting from a greater supply of cooled water from more efficient plants, the energy-recovery possibilities due to incorporating the solar desiccant dehumidification system in a conventional air conditioning system are significant for the hot, humid weather in Hong Kong. The hybrid system reduces annual service energy consumption by 6760 kW/h [18].

The use of a plant intended to function at peak efficiency to produce the best water with the least amount of electricity was presented as a way to extract water from atmospheric air. The study also looked into whether solar energy can be employed in atmospheric water generators (AWGs) as a new source of fresh water. Using HOMER software, a recommended solar AWG unit was conceived, assessed, and designed. The studies revealed that the produced water is safe to drink and that solar-powered AWG technology is technically possible. AWG technology generates consumable water at a lower cost than bottled water and is more environmentally friendly [19]. This study described the computing methods and component selection for significant cooling system components used in the atmospheric generator instead of a condenser to increase the volume of received water. Thermoelectric refrigerating machines (TRMs), vapor-compression refrigerating machines (VCRMs), and Stirling-cycle refrigerating machines (SRMs) have all been studied [20].

AWG technology provides drinkable water at a competitive price compared to bottled water, which is more environmentally and health-friendly. In the program (Cool Pack), return Rankine cycles were established (VCRM). The job required the use of two refrigerant coolants: R134a and R502. Cycle requirements were calculated for autumn, spring, summer, and winter. The results revealed that the optimal mode occurs in autumn, and the optimal refrigerating machine for work in fixed settings with the highest water productivity per day is a VCRM. Vinay et al. developed a water generator that was used to generate water by extracting humidity from atmospheric air. The system's design is simple, and it consists of a condenser, evaporator, compressor, blower, and copper tube. Its operation is similar to that of refrigerators and air conditioners in that ambient air is converted to pressurized air via the compressor and then transferred to the condenser tubes to cool the dew point, transforming into a water droplet. Several filters, including carbon, reverse osmosis, water filter, UV sterilization, and light sand, are housed in a tank [21].

This project was based on refrigeration, which is defined as the process of moving heat from one point to another. A compressor, condenser, expansion valve, and evaporator

are employed, as shown in Figure 3 [22]. This experiment was carried out using two or three refrigerants: R134a, R290, and HC R290. It was discovered that the system produced good economic outcomes with HC R290. As a result, HC R290 was deemed to be the best refrigerant for this job. In 1.5 h, the amount of water collected was 250 mL [23].

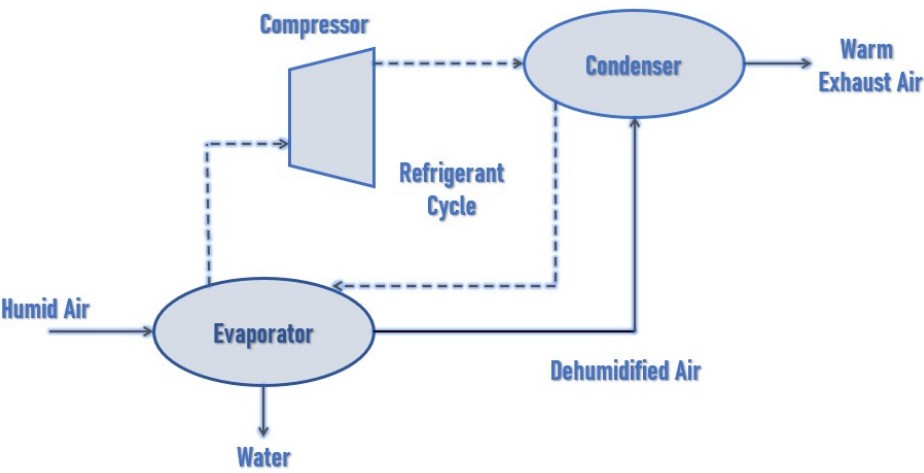

**Figure 3.** Block diagram of water extractor [22].

### 2.2. Earth-Water Collector

An arrangement consists of columnar and slanted channels over the Earth's surface that are used to obtain pure water from atmospheric air by cooling humid atmospheric air to a temperature lower than the dew point [7,24]. The amount of gaseous solar radiation absorbers in the atmosphere, such as water vapor and ozone, can also be significant. Downward thermal emission from the atmosphere, which is influenced by clouds and water vapor, as well as upward thermal emission, which is lowered by upward thermal emission, make up the total net radiation that heats the terrestrial surface. Because this 'Earth' radiation is more weakly dependent on surface temperature than energy fluxes from surface evaporation and dry sensible heat, we use the phrase 'total net radiation' to describe the atmospheric radiative fluxes to the surface in the next section [25].

When the brightness of the sun increases, the Earth's surface becomes dry. The dry surface's depth differs based on the amount of average rainfall, the depth of the capillary motion, and the soil type in that location. Inside this dry surface, a moist coating is present. This happens due to the interaction between capillary movement and underground water. This capillary action means that this water is lifted to the Earth's surface by minute slits in the soil. When the sunlight heats the upper surface using solar energy, this water disperses into vapor [26]. A four-sided figure with a glaze at the slope called the Earth-Water Collector is used (Figure 4). When the upper surface of the Earth is heated by the sun in the form of solar energy, water is evaporated from the surface through water vapor [27]. This process is known as convection. This collection of water then reaches the ground surface due to gravity as portable fresh water.

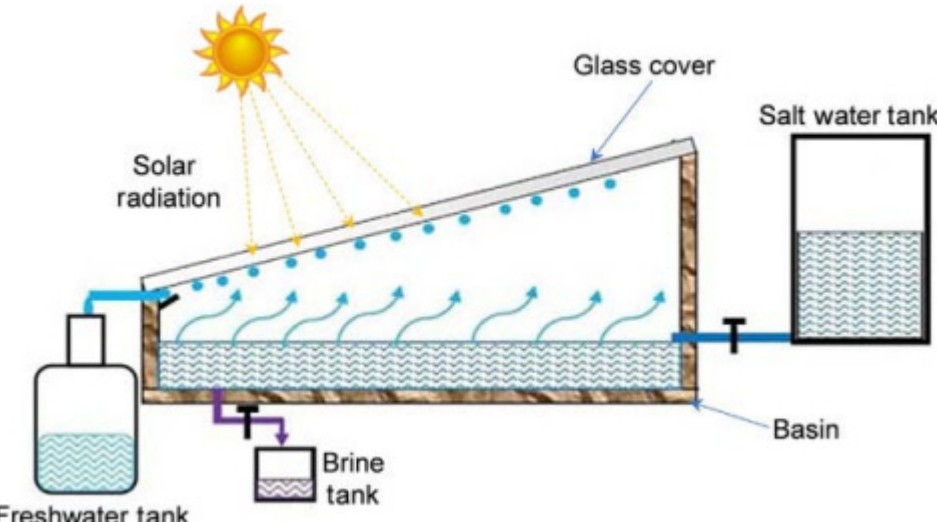

**Figure 4.** Principle of Earth-Water Collector (adapted with permission from Reference [28], Copyright 2015, Elsevier).

### 2.3. Absorption–Regeneration Cycle

A conceptual cycle was developed to describe water vapor absorption from atmospheric air, followed by the regeneration process. $CaCl_2$ was used as a desiccant in this study. The ambient conditions' influence on the cycle's operating limits was predicted [29]. The absorption–regeneration cycle is shown in Figure 5. These are the four thermal processes:

(a)     Process 1_2: isothermal water vapor absorption from the air;
(b)     Process 2_3: heating the absorbent at a steady concentration;
(c)     Process 3_4: constant pressure absorbent regeneration;
(d)     Process 4_1: cooling the absorbent at a steady concentration.

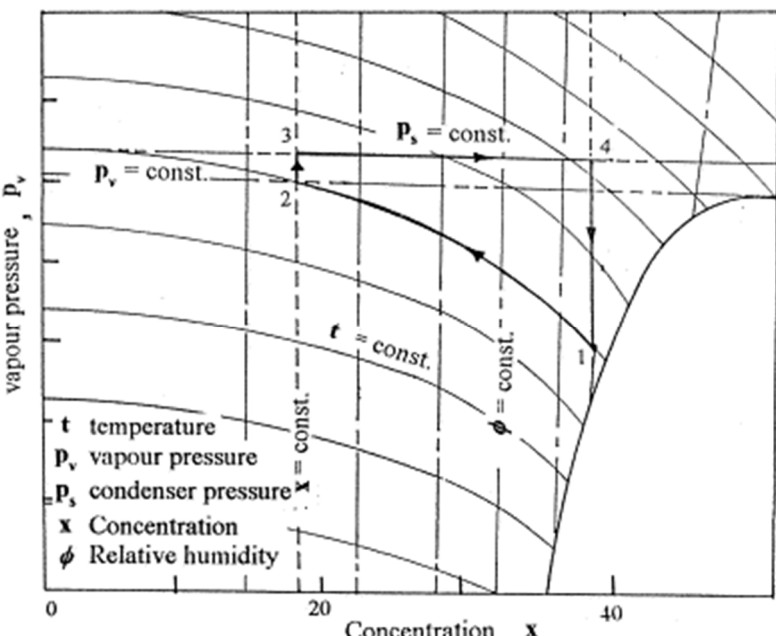

**Figure 5.** The absorption–regeneration cycle (adapted with permission from Reference [29], Copyright 2000, Elsevier).

According to theoretical studies, the limits of weak and strong solution concentrations considerably impact the cycle efficiency value. A cycle efficiency of more than 90% can be

achieved for various values of a high solution concentration; however, this value drops when the variation among the strong and weak solution concentrations is minimal. The absorption–regeneration cycle is shown in Figure 5.

Sultan et al. proposed an unconventional practice of steam extraction from atmospheric air on a 24 h basis using a tight structure. The approach is based on the absorption–regeneration process with forced convection in a filled tower with two comparable columns, each filled with a similar bed made up of upright multi-layers of a fabric medium soaked with $CaCl_2$ solution of varying concentrations. Based on the experimental findings, an analytical model was created [30]. The highest concentration of the desiccant was discovered to be dependent on the primary concentration, penetration, and regeneration temperatures. The system's efficiency increased as the primary concentration increased, and it decreased as the absorption temperature and air velocity employed in the regeneration process increased [31].

*2.4. Dew Collection*

A plan for large-scale dew collection as a source of fresh water was presented. Using four plastic pipes, the system involves delivering cold seawater (5 °C) from a depth of approximately 500 m and around 5 km from the shore. It then flows through a heat exchanger, condensing 643 m$^3$ of dew in 24 h. Three wind turbines (each with a capacity of 200 kW) pump seawater from the sea into the area. The system's technical and economic feasibility was examined. It was determined that the current concept is not economically viable compared to a reverse osmosis facility of comparable capacity [32]. Dew is formed when moisture in the air condenses and is utilized for drinking and irrigation. Dew-collecting surfaces can benefit from the radiative cooling capabilities of polymer foils. This research was focused on dew development on colored polyethylene foils that have been radioactively cooled. The findings revealed that the volume of collected dew per m$^2$ is minimal, but large-scale systems can be inexpensive [33,34]. Dew collection will play an important role in the regions of our planet that are arid [35].

Experiments have shown that 0.22 L/day of water can be collected in a single night of operation, with model predictions and experimental findings agreeing well [36]. The DEW project began by installing a 15 m$^2$ roof with a commercial plastic cover chosen for its exceptional dew-gathering qualities on the tiny Mediterranean islands of Bisevo, Croatia. Rain and dew water measurements were taken for several years, and the results will be compared to meteorological data gathered on the site. According to the previous measure, the total collected water was 642 L. The volume of dew water was 222 L, which was equal to 26% of all collected water (21 April 2005–21 October 2005) [37]; the study evaluated the feasibility of using fog collectors to obtain water for various applications by the local population (domestic, agriculture, cattle raising, and forestry).

There are three types of fog collectors: (a) a flat rectangular fog collector, (b) a round rectangular fog collector, and (c) a simple fog collector in the form of a cylinder. A fog collector with two cylinders was put to use. The results revealed that when a larger amount of water was collected on the double-cylinder fog collector, it was possible to achieve an average water production of 0.53 L/m2/day and a maximum water collection of 3.3 L/m$^2$/day [38].

Figure 6 shows a flat dew collector and overturned pyramid collector. Experiments to gather passive dew were conducted in a grassland setting. A 1 m$^2$ flat collector with a 0.39 mm polyethylene foil covering angled at a 30° angle and a second dew collector in the shape of an overturned pyramid were erected. Two models were employed to predict dew collection on the two surfaces: one was a simple surface energy budget model, while the other was an aerodynamic model. According to the statistics, the daily average dew was 0.12–0.03 mm on the grass cover, 0.1–0.06 mm in the flat collector, and 0.15–0.05 mm in the overturned pyramid collector. A fog-water collector system is shown in Figure 7 [39]. The overturned pyramid produced more dew than the flat collector or grass cover due to improved insulation and a lower sky observation point. The study assumed that water can

be extracted from fog under specific topographical and climatic conditions. When the fog's microscopic droplets encounter a solid surface, they precipitate. Fog might be regarded as an alternative source of fresh water when it occurs regularly. Three standard fog collectors (SFCs) were built and tested at three different altitudes (22, 603 and 200 m). According to the findings, the amount of water collected was 6.125 L/m$^2$/day at the highest altitude and 3.3 L/m$^2$/day at the lowest height [40].

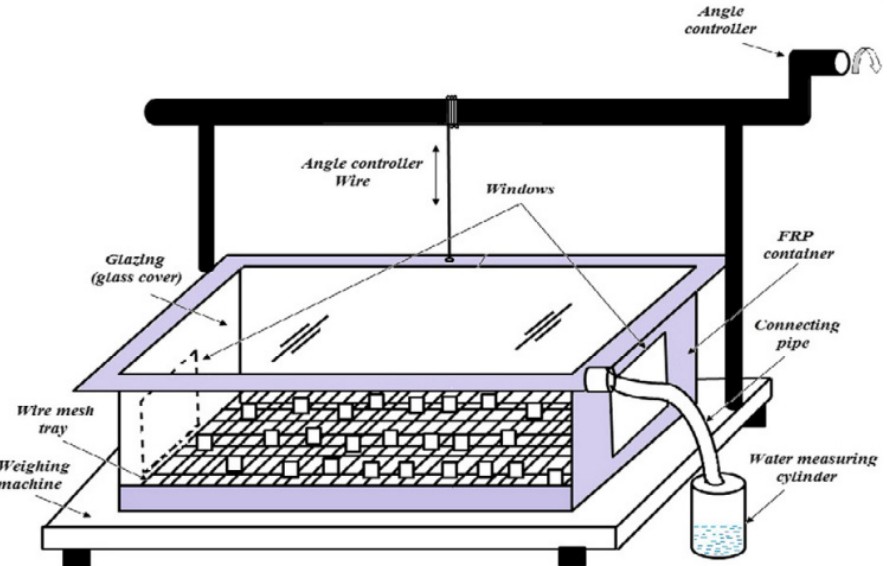

**Figure 6.** Flat dew collector and overturned pyramid collector.(Adapted with permission from Reference [35], Copyright 2015, Elsevier).

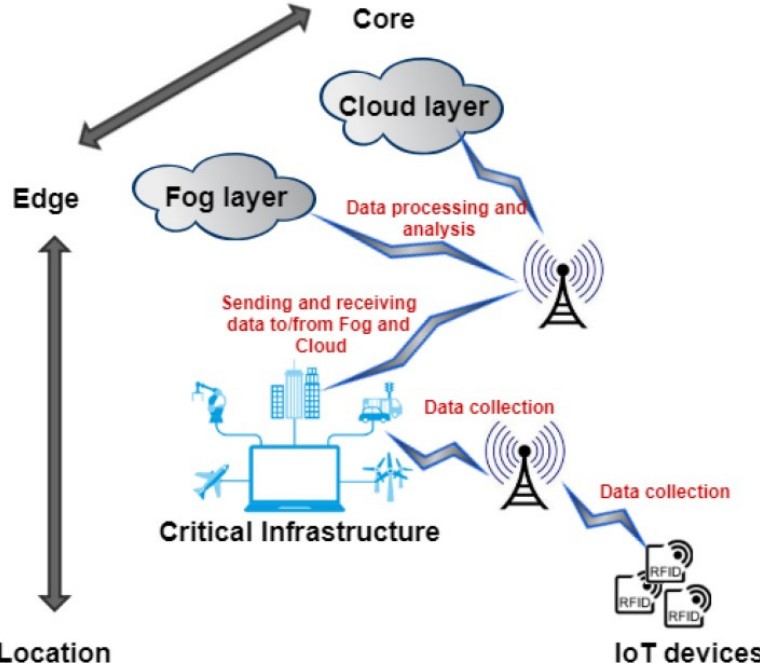

**Figure 7.** Fog-water collector system (reprinted from Reference [39] under the terms of the Creative Commons Attribution License CC BY 4.0 license, https://creativecommons.org/licenses/by/4.0/ (accessed on 25 October 2021)).

*2.5. Desiccant Systems*

Water is extracted from humid air using desiccant materials. A desiccant with a high concentration can absorb moist air at night (absorption phase), and the absorbed water can

be removed from the diluted desiccant during the regeneration process by heating. When employing the desiccant method, the regeneration process can be carried out at relatively low temperatures, ranging from 40 to 70 °C, which is ideal for solar stills [41]. The primary benefit of solar-powered water extraction is its purity, sustainability, and self-sufficiency, as it does not require any infrastructure [42].

Using ethylene glycol as a liquid desiccant, Hall et al. developed an absorption method for obtaining fresh water from ambient air, followed by recovery in a solar still [43]. The water retrieved with this method was affected by temperature and humidity. The data were presented in a composition psychometric chart, although the investigation lacked information on the mass of the collected water [44].

An unorthodox method that captures water from the atmosphere using a solid desiccant was developed. It was also examined whether humid air could be collected by cooling it to a temperature below the dew point using an air-conditioning system [45]. In a method for extracting humidity from atmospheric air, a desiccant pond was used to absorb humidity from the air and produce a water-rich desiccant. The sun heats the water-rich desiccant, causing the moisture to evaporate and mix with the ambient air, raising the ambient air's dew point, condensing the vaporized moisture to generate a drinking water condensate, and returning the water-poor desiccant to the desiccant pond. An S-shaped composite material was employed to manufacture water from atmospheric air by soaking it in a physiochemical liquid to absorb humidity from the air [46]. It was discovered that when the temperature rose, the absorptivity of the surface fell, necessitating a 50-degree angle to absorb humidity. One liter of water might be produced per square meter of composite material.

The amount of water absorbed by the desiccant was calculated from the ambient air in the absorber as a function of climate data and the desiccant's initial conditions. The liquid desiccant was calcium chloride, and the absorptivity surface was an inclined surface exposed to the environment. The desiccant runs along the surface, absorbing humidity as it does so because of the difference in vapor pressure between the air and the desiccant surface [47]. The sun's rays heat the water-rich desiccant, forcing moisture out and condensing the vapor stream. The results showed that increasing the desiccant flow rate, air relative humidity, and wind speed increased the mass of absorbed water.

Gandhidasan and Abulhamayel et al. suggested that water be extracted from the atmosphere using a suitable liquid desiccant. Using the same unit, the process has a nighttime component where humidity is absorbed and a daily part where humidity is desorbed. Figure 6 depicts the system's structure, consisting of a flat, blackened, slanted surface covered with single glazing with a 45 cm air gap. The desiccant flows as a thin layer over the glazing at night, and the desiccant is diluted because of moisture absorption. During the day, solar energy heats the diluted desiccant as it flows downward and across the absorption surface. Moisture vaporizes in desiccant vaporizers and condenses on the underside of the glazing. A total of 1.92 kg/m of water was generated [48].

A new type of composite material known as selective water sorbents (SWSs) was created for removing humidity from the air. As host materials, silica gel ($SiO_2$), alumina (IK 02 200), and porous carbon (subunit) were utilized. Lithium bromide (LiBr) and calcium chloride ($CaCl_2$) were utilized as hygroscopic salts. The ability to produce 3–5 tonnes of water/10 tonnes of dry SWS was demonstrated in lab experiments [49].

For water production from atmospheric air, Gad [44] used an integrated (solar/desiccant) device. At night, the process absorbs humidity from the air, regenerates the desiccant, and condenses the vapor during the day. To improve the mass transfer area, a thick, corrugated textile layer impregnated with $CaCl_2$ was utilized as a bed. At night, a fan was used to circulate air over the bed. In addition, a condensing device was used. The operation was repeated twice, once without the condenser and once with it. The results revealed that employing the condenser device reduced the system's efficiency by 5%. In total, 1.5 $L/m^2$/day was produced. Hamed et al. used a horizontal sandy bed saturated with $CaCl_2$ to investigate the natural absorption of water vapor from a gaseous-air combination.

There were seven levels in total, each with a different desiccant/sand ratio ranging from 0% to 100% (0.1 to 0.4). The effects of the desiccant concentration and Grashof number on the mass transfer coefficient and mixing ratio (desiccant/sand) on the absorption rate were investigated. The data demonstrated that as the concentration increased, the mass transfer increased as well. Furthermore, it was discovered that mass transfer decreased significantly as the mixing ratio was reduced [50]. A schematic of fresh water from the humid atmosphere unit is shown in Figure 8.

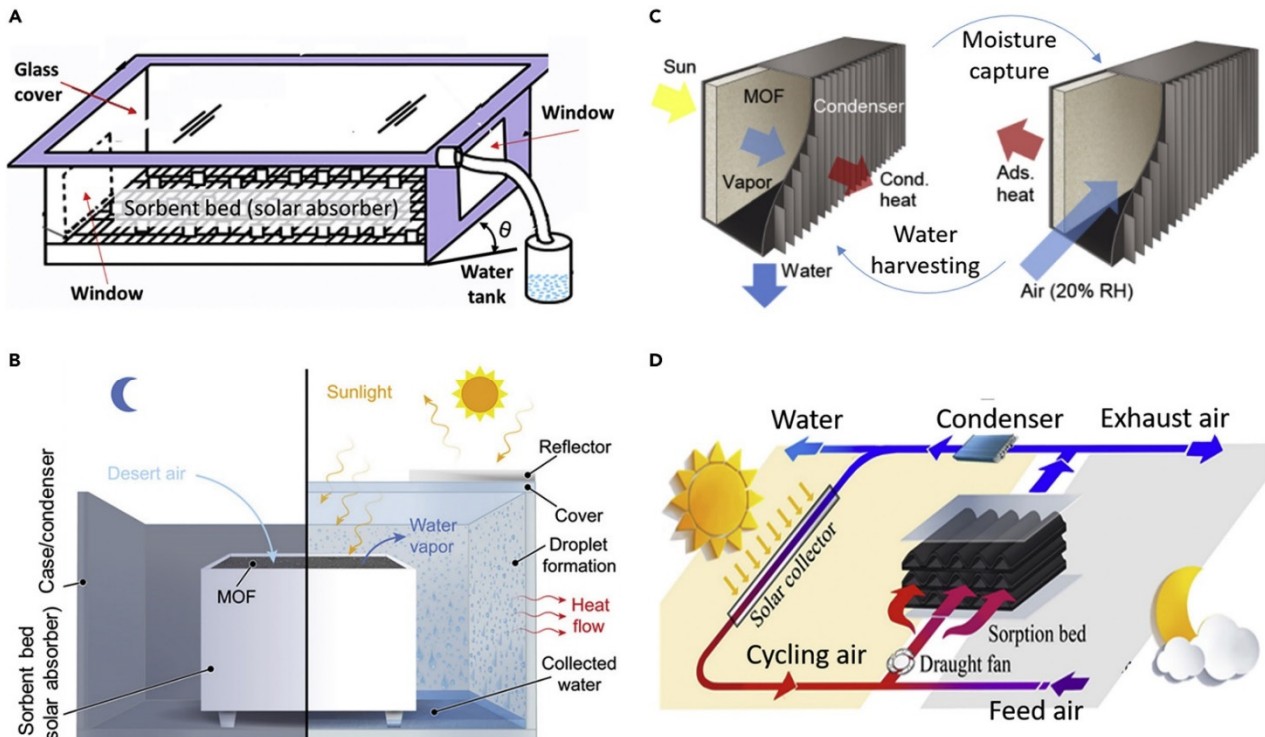

**Figure 8.** Schematic of fresh water from the humid atmosphere unit. (**A**,**B**) Glass-covered greenhouse absorber; (**C**) sandwich plate absorber; (**D**) packed column absorber (adapted with permission from Reference [51], Copyright 2018, Elsevier).

Kabeel et al. developed a method to extract water from the atmosphere. A system consisting of a solar collector unit including a sandy bed saturated with $CaCl_2$ was used. Sand was used to replicate the conditions observed in Arab deserts. Three different tilt degrees (15°, 20°, and 25°) were used to test the system. A theoretical model was used to investigate the effects of various parameters, such as solution concentration and solar radiation intensity. The results showed that it could produce 1.2 L/m²/day of pure water, with the best results obtained with a tilt angle of 25°. Scrivani et al. looked into non-heating and cooling applications for the solar trough concentration, such as water generation from atmospheric air. This was accomplished using a solar concentrator and a double-effect ammonia chiller. It has the advantage of condensing air humidity using chillers using the same energy source that heats the air: solar heat. As such, only two pieces of solar-powered equipment are required, both of which require no additional energy sources other than the troughs' orientation and the diathermic fluid delivered via the collector [52].

Two identical pyramid-shaped multi-shelf solar systems aid in the extraction of water from the atmosphere. The two systems have the same proportions, but the types of beds are different. The beds are impregnated with calcium chloride. At night, all four glass faces of the pyramid are opened to absorb moisture from the air near the beds. Then, the faces are closed to extract water from the air. The first pyramid's bed is made of saw wood, while the second is made of cloth with the same dimensions. Figure 9 shows a

solar desiccant/collector system for water recovery [53], and Figure 10 shows a pyramid quadratic solar system. Tests were conducted in various climates to see how the pyramid design affected absorption [54].

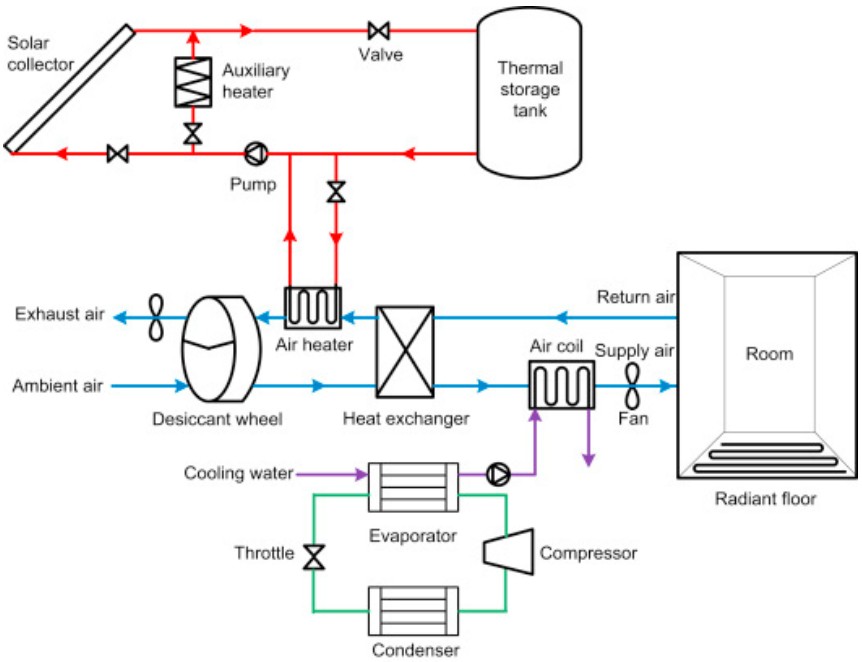

**Figure 9.** Solar desiccant/collector system for water recovery (adapted with permission from Reference [53], Copyright 2014, Elsevier).

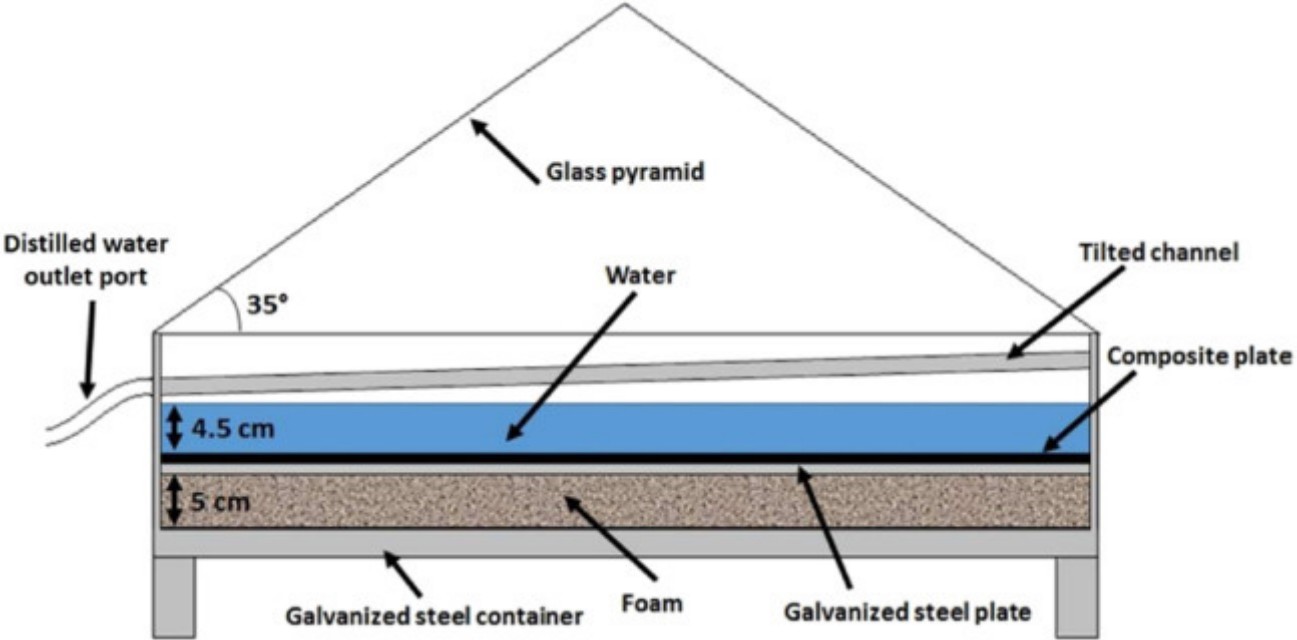

**Figure 10.** A pyramid quadratic solar system (adapted with permission from Reference [55], Copyright 2017, Elsevier).

According to the findings, the garment bed absorbs a larger volume of solutions than the cut wood. A new high-efficiency composite adsorbent for extracting water from the atmosphere using solar energy was created with a rate of 2.1 L/m²/day. The composite material comprises a host matrix of crystalline material with large pores (MCM-41) and calcium chloride as a hygroscopic salt [56]. The findings demonstrated that the composite

material has a high adsorption capacity (1.75 kg/kg desiccant), a unique adsorption rate, and a low-temperature regeneration process. A large amount of water (1.2 kg/m$^2$) was produced. Water was created at a rate of 1.2 kg/m$^2$ [57].

The principal application of solar concentration for collecting portable water from the atmospheric air is described in this study. The findings of the AQUASOLIS study were used to assess the usefulness of solar trough concentration plants for uses other than heating and cooling, particularly water generation for a variety of applications. Abulhamayel and Gandhidasan et al. observed an absorber/desorber that was built and found it to be suitable for gathering the moisture that abounds in the humid atmosphere [58].

Calcium chloride was employed as a functioning desiccant. The desiccant solution passes over the absorber and is exposed to the ambient air at night, removing humidity from the air. After covering the absorber with a glass cover throughout the day, the diluted solution runs over it again. Four different desiccant flow rates and concentrations of desiccant were used in the trials (32.5–33.5 percent). The results revealed that increased ambient air temperature, wind speed, and solution concentration and a reduced desiccant flow rate enhance the absorption efficiency.

In contrast, increased solar radiation and ambient air temperature and a decreased desiccant flow rate increased the desorption rate. Depending on the desiccant flow rate, the water absorption rate was 2.11 L/m$^2$/day, while the water desorption rate was 1.15 L/m$^2$/day. Because it uses solar-powered thermoelectric generators, the technology is appropriate for Arab Gulf countries or regions with comparable water conditions. Using Star-CCM+, a simulation was run. The parameters (pressure drop across the flow channel, water productivity/m$^2$, impact of ambient air temperature, and humidity) were assessed (the Red Sea, Arabian Gulf, and South Spain). The amount of water produced rose to 3.9 L/h/m$^2$ [59], while the amount of pumping power required for the air fan remained the same (9.1 W) [4]. Investigations on new composite material for the storage and production of water from ambient air were conducted. Three different solar glass box-type systems (SGDBSs) were used with a collection area of 0.36 m$^2$. Six different concentrations of $CaCl_2$ composite were used in the tests. The amount of absorbed water at night, the temperature of the composite material, the temperature of the device's internal space, and the amount of water generated during the day all rose as the desiccant concentration rose. A composite material with a percentage of 60% produced the most water (180 mL/kg/day).

Many experiments were conducted in this study to determine the capability of a newly created composite desiccant material ($CaCl^2$/flower foam) for extracting water from atmospheric air. The floral foam was employed as the host material, and $CaCl_2$ was used as the hygroscopic salt. Three different sets of SGDBs were employed. Six distinct concentrations of $CaCl_2$ composite were created. The greatest adsorption rate (0.043992 kg/h) and water production (0.35 mL/cm$^3$/day) were discovered at a concentration of 37 percent $CaCl_2$. The amount of water produced increased as the concentration of $CaCl_2$ rose [60].

The air gap height, inclination angle, effective glass thickness, and sufficient glazing number were investigated as design parameters for collecting drinkable water from the air utilizing silica gel ($SiO_2$) as a solid desiccant in the study. There were three different numbers of SGBDs used [61]. The results revealed that the design factors for maximum production were the air gap height (0.22 m), inclination angle (30°), the effective thickness of glass (3 mm), and single glazing. Most of the water was produced (200 mL/kg/day) during the trial [62]. William et al. designed and built a trapezoidal prism collector with four fiberglass faces. The collector has a multi-shelf bed on the inside to enhance the bed surface area. As a bed, sand and fabric layers were used, with $CaCl_2$ at a 30% concentration acting as a desiccant. At a $CaCl_2$ concentration of 30%, evaporated water for cloth and sand can reach 2.32 slit/m$^2$/day and 1.23 slit/m$^2$/day, respectively. The efficiency of the cloth and sand systems was 29.3 percent and 17.76 percent, respectively. Various strategies that have been published for extracting water from ambient air have been given in Table 1.

**Table 1.** Various strategies that have been published for extracting water from ambient air.

| Reference No. | Bed Type | Desiccant Type | Production Amount | Test Location | Test Date |
|---|---|---|---|---|---|
| [43] | Sheet of plywood | Ethylene glycol | - | Manhattan, USA | 1966 |
| [46] | S-shaped composite material | Physicochemical adsorption | $1 \text{ L/m}^2$ | Holland | 1987 |
| [45] | - | Triethylene glycol | - | USA | 1993 |
| [47] | - | $CaCl_2$ | - | Dhahran, KSA | 1996 |
| [48] | - | - | $1.92 \text{ L/m}^2$ | Dhahran, KSA | 1997 |
| [49] | $SiO_2$ or $Al_2O_3$ or C | $CaCl_2$ or LiBr | 3–5 ton/ton of dry $SWS_s$ | Novosibirsk, Russia | 1998 |
| [44] | Thick corrugated layer of cloth | $CaCl_2$ | $1.5 \text{ L/m}^2$ | Egypt | 2000 |
| [50] | Sand | $CaCl_2$ | - | Egypt | 2002 |
| [8] | Sand | $CaCl_2$ | $1.2 \text{ L/m}^2$ | Egypt | 2004 |
| [54] | Saw wood and corrugated cloth layer | $CaCl_2$ | $2.5 \text{ L/m}^2$ | Egypt | 2007 |
| [32] | MCM-41 material | $CaCl_2$ | $1.2 \text{ L/m}^2$ | Shanghai, China | 2007 |
| [56] | - | $CaCl_2$ | Absorption: $2.11 \text{ L/m}^2$ Desorption: $1.15 \text{ L/m}^2$ | Dhahran, KSA | 2010 |
| [24] | Sand | $CaCl_2$ | $1 \text{ L/m}^2$ | Taif city, KSA | 2011 |
| [59] | - | $CaCl_2$ | $3.9 \text{ L/h/m}^2$ | Tanta, Egypt | 2014 |
| [62] | Sand and cloth layer | - | $1.23 \text{ slit/m}^2/\text{d}$ $2.32 \text{ slit/m}^2/\text{d}$ | Cairo, Egypt | 2015 |
| [4] | Saw wood | $CaCl_2$ | 180 mL/kg/d | Kurukshetra, India | 2015 |
| [60] | Floral foam | $CaCl_2$ | $0.35 \text{ mL/cm}^3/\text{d}$ | Kurukshetra, India | 2015 |
| [63] | Vermiculite + Saw wood | $CaCl_2$ | 195 mL/kg/d | Kurukshetra, India | 2015 |
| [64] | Active carbon felt + Nano silica grains | LiCl | 14.7 kg/kg of water | Shanghai, China | 2017 |
| [65] | Cloth layer | $CaCl_2$ | 0.3295–0.6310 $\text{kg/m}^2/\text{d}$ | Mansoura, Egypt | 2018 |
| [66] | Sand | CM1: LiCl CM2: $CaCl_2$ CM3: LiBr | CM1: 90 mL/d CM2: 115 mL/d CM3: 73 mL/d | Kurukshetra, India | 2018 |

Kumar et al. developed and tested a new composite desiccant material, $CaCl_2$–vermiculite saw wood, for collecting water from ambient air. The hygroscopic salt was $CaCl_2$, and the host material was vermiculite saw wood. Six different ethanol concentrations were made using a solid desiccant type device with a collector area of $0.36 \text{ m}^2$ and a $30°$ inclined inclination ($CaCl_2$). The adsorption and desorption rates were highest when the adsorbent and desorbing concentrations were highest ($CaCl_2$). The highest amount of generated water was $500 \text{ mL/m}^2/\text{day}$ when using 2.5 kg of composite desiccant. The most effective attribute for water creation was the desiccant content. Wang et al. proposed the construction of a highly efficient semi-open system for the production of potable water. The composite material's unit structure (active carbon felt (ACF) with nanosilica grains aggregated in pure ACF fibers gap soaked with lithium chloride) was made into a corrugated shape that can be easily constructed as mass transfer channels. The process is divided into absorption at night and desorption during the day [64]. The sorbent unit was investigated (CFX). The findings revealed that the amount of absorbed water grew

as relative humidity rose in the absorption phase, while the amount of released water decreased in the desorption process. As the temperature of desorption rose, the amount of water in the system increased. Overall, 40.8 kg of composite material was used to collect 14.7 kg of water. Talaat et al. looked into the factors that influence the performance of a finned, portable, lightweight device that uses $CaCl_2$ as a desiccant to collect fresh water from the atmosphere. A finned double-faced conical absorber was manufactured and used to capture moisture from the ambient air at night. During the day, a transparent double-faced conical cover surface was used to capture heat. The system's productivity ranged from 0.3295 to 0.6310 kg per square meter per day, with the produced water costing 0.062 USD per kilogram [65]. Figure 11 shows the test rig operation during the daytime, and Figure 12 shows a schematic diagram of the main parts and thermocouples' distribution on the absorber and cover.

Srivastava et al. used three different composite materials to extract water from atmospheric air: LiCl/sand (CM 1), $CaCl_2$/sand (CM 2), and LiBr/sand (CM 3), each with a 37 percent concentration of hygroscopic salt and sand as a host medium. Absorption and desorption mechanisms were used to remove water from the air. A new Scheffler reflector (1.54 m$^2$) was used during the day to refresh the area [66]. Figure 13 shows a daytime schematic diagram of water generation employing a composite desiccant material and a fixed focus concentrator. Table 1 shows various strategies that have been published for extracting water from ambient air.

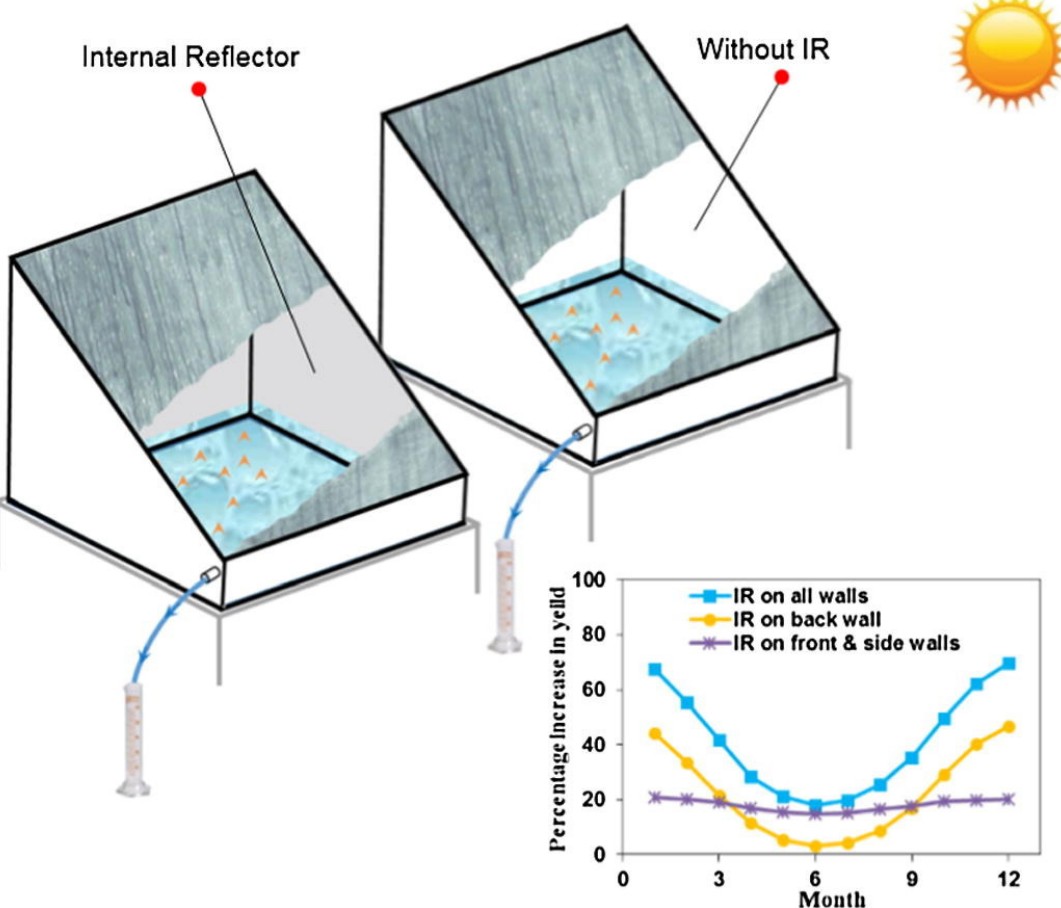

**Figure 11.** Test rig operation during daytime (adapted with permission from Reference [67], Copyright 2016, Elsevier).

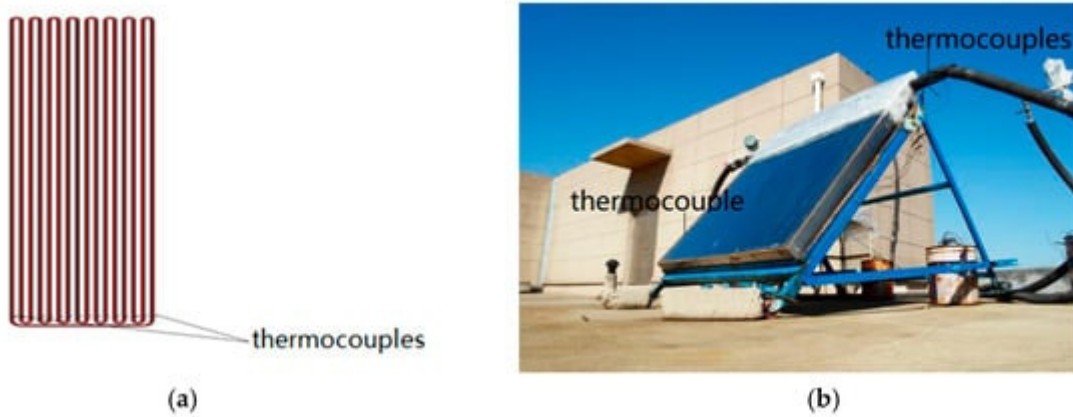

**Figure 12.** Schematic diagram of main parts and thermocouples' (**a**) distribution on the absorber and cover (**b**) (reprinted from Reference [68] under the terms of the Creative Commons Attribution License CC BY 4.0 license, https://creativecommons.org/licenses/by/4.0/ (accessed on 25 October 2021)).

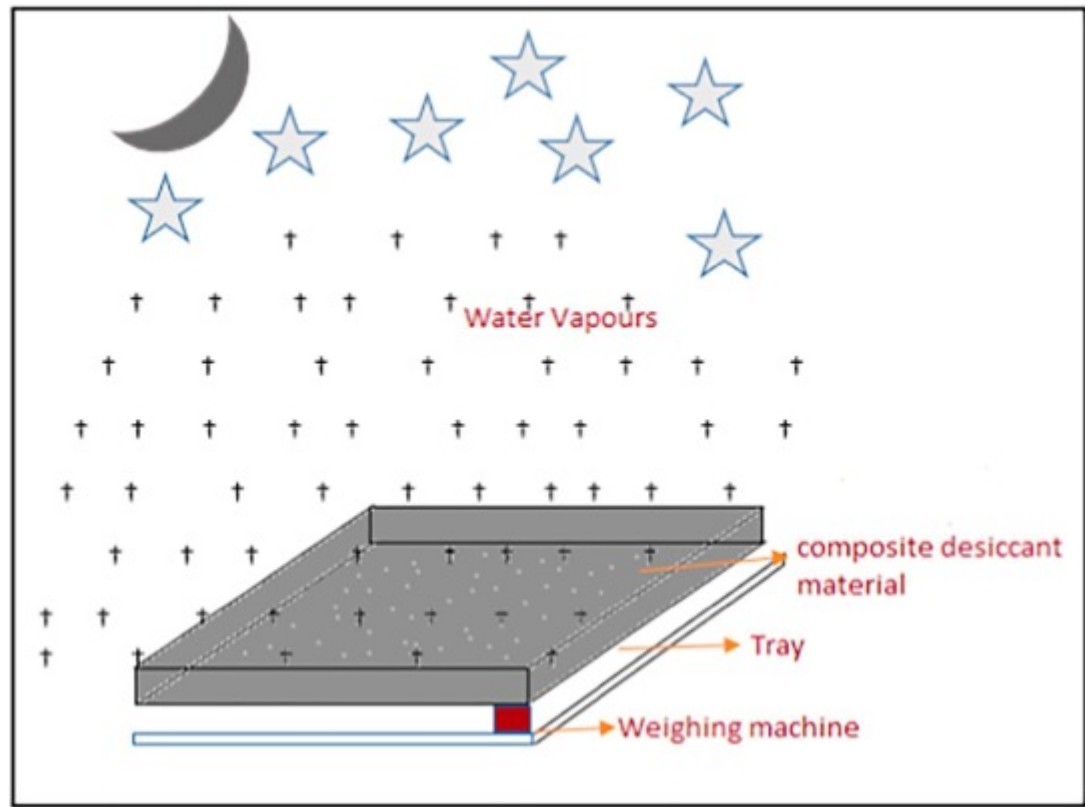

**Figure 13.** Schematic diagram of water generation by composite desiccant material using fixed focus concentrator during the day (adapted with permission from Reference [66], Copyright 2018, Elsevier).

An experimental work was conducted in Malaysia's tropical outdoor environment conditions, in which the performance of a medium-scale atmosphere water generator (MSAWG) system was tested for a 48 h period from 8 p.m. on 18 March 2020 to 8 p.m. on 20 March 2020. It is observed in Figure 14 that the MSAWG system's hourly water production climbed from 110 mL/h at 9 p.m. (night) on 18 March 2020 to 204 mL/h at 5 a.m. (early morning) on 19 March 2020 as the corresponding relative humidity rates increased from 62 to 80 percent. Hence, AWG can be used as a long-term solution to consistently generate daily fresh water, helping to alleviate the country's unpredicted water crisis [69].

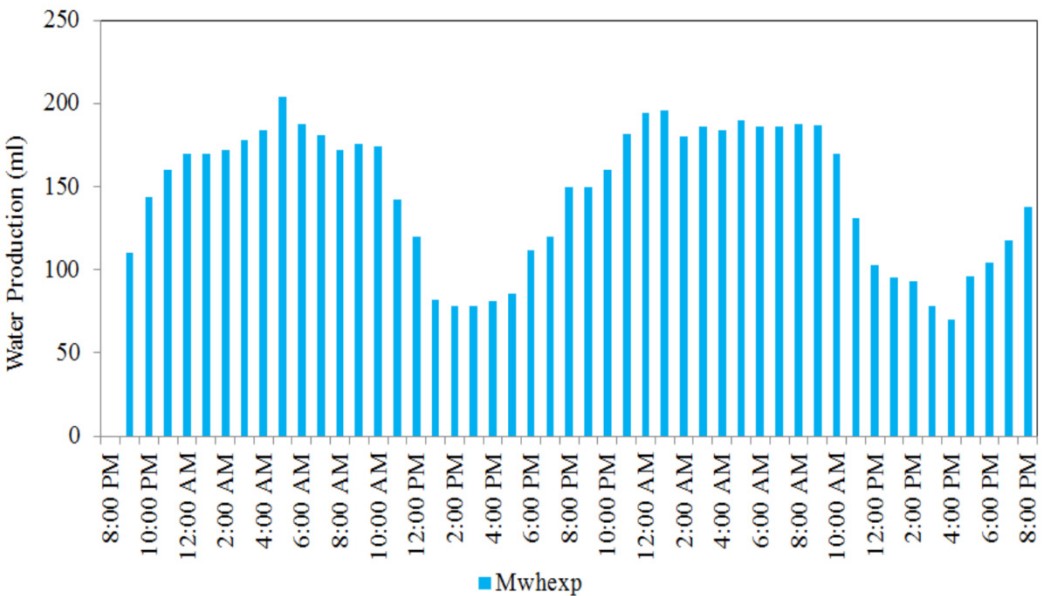

**Figure 14.** Values of hourly water production of MSAWG in a 48 h period from 8.00 p.m. on 18 March 2020 to 8.00 p.m. on 20 March 2020 (reprinted from Reference [69] under the terms of the Creative Commons Attribution License CC BY 4.0 license, https://creativecommons.org/licenses/by/4.0/ (accessed on 25 October 2021)).

## 3. Policy Perspective

Environmental and water resource economic debates are all new topics driving policy decisions on this technology. The best support for AWG system adoption reveals interesting choices, comparable to those seen in intermittent renewable energy sources. On the one hand, a subsidy for the purchase of an AWG or for the price of power may result in social benefits such as cheaper investment in water infrastructure, less environmental harm, and lower health risks. High AWG penetration, on the other hand, may raise overall energy consumption, impair the economy of scale for alternate water resources, and incur economic costs [70].

## 4. Conclusions

Even though there are sufficient water resources, their uneven circulation leads to a shortage and requirement for portable fresh water. More than two billion people live in water-stressed areas. Studying a range of research shows that the most widespread method for actively extracting water from atmospheric air is the air-cooling method, where water is produced for free because it is a byproduct. In contrast, the best method for extracting water passively (without using an energy source) is the desiccant method because it can be used anywhere, unlike the dew collection method, which can only be used in regions where very high quantities of moisture occur, such as coastal and mountain areas, and in contrast to Earth-water collecting methods, which can only be used in open, sandy regions such as deserts. The maximum amount of water collected is 2.5 L/m$^2$/day using $CaCl_2$ as a desiccant, a pyramid-shaped multi-shelf device, and solar energy as a heating source. Researchers have begun to use the desiccant method for water extraction because it has a high production rate, does not require an energy source, and uses solar energy as a heating source. This research work will support further research on the extraction of water from atmospheric air in arid zones for the benefit of society

## 5. Future Work and Limitations

A water extraction system is a potential way to rapidly supply drinking water. The ability to connect an extraction system to a local Wi-Fi or Bluetooth device, as well as its ability to be an Internet of Things (IoT) device, will allow the user to engage with

it intuitively. Water extraction systems should be further developed in the near future. Water extraction is capable of expanding the limitations of present technologies and will considerably help to reduce water shortages as a result of all of the aforementioned research efforts and the attractive possibilities that they have offered.

**Author Contributions:** Y.W.: Conceptualization, methodology, investigation, writing—original draft preparation, funding acquisition S.H.D.: writing—original draft preparation, resources; H.A.Z.A.-b.: writing—review and editing, supervision; D.V.; writing—review and editing, supervision; F.W.: supervision, resources. All authors have read and agreed to the published version of the manuscript.

**Funding:** This research was funded by "Jiangsu Propaganda culture development special project—think tank research topic", grant number CJ20028, and "Jiangsu University Humanities and Social Sciences Out-of-school Research Base Project "Tonghu Industrial Cooperative Development Research Base", grant number 2020THKFKT04.

**Conflicts of Interest:** The authors declare no conflict of interest.

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
