# Peer review of "A Recent and Systematic Review on Water Extraction from the Atmosphere for Arid Zones"

_energies, doi:10.3390/en15020421_

Round 1
Reviewer 1 Report
Dear Authors, thank you for the interesting and useful study.
The structure of the article is well verified, contains the sections necessary for the review manuscript, although the section “Results and Discussion” is absent. The introduction is written rather succinctly, little attention is paid to the relevance of research. The list of used literature contains 68 sources, only 23 of which - over the past 5 years. Some figures require quality improvement.
Below are the main comments to the manuscript, the remaining technical comments are given in the text of the Reviewer’s PDF.
- Comparison of the economic effects of the technologies under consideration is not provided.
- There is no comparison of the productivity of the technologies under consideration. A detailed comparison of productivity (l / m2 / day or kg / m2 / day) should be given in the Results and Discussion section.
- There is a very lack of specifics (an example of the application of technology with an indication of the country and city).
- Table 1 contains a comparison of all considered methods? If so, why is it listed in the “Desiccant Systems” section? I recommend that you pay attention to its detailed consideration in the "Results and Discussion" section.
- I recommend adding a section "Acknowledgments".
- The "Conclusions" section should be expanded. I recommend adding to it a comparison of specific indicators of all the technologies considered.

Author Response
The authors would like to place their sincere thankfulness to the Editor, Reviewers and Coordinators of the Energies for considering our paper for its publication and further providing their valuable suggestions.
The answers for the queries raised by the reviewers are answered herewith clarity and humbleness.
Comments to the Author
Reviewer 1:
The structure of the article is well verified, contains the sections necessary for the review manuscript, although the section “Results and Discussion” is absent. The introduction is written rather succinctly, little attention is paid to the relevance of research. The list of used literature contains 68 sources, only 23 of which - over the past 5 years. Some figures require quality improvement.
As per the reviewer comments, recent articles have been added.
Below are the main comments to the manuscript, the remaining technical comments are given in the text of the Reviewer’s PDF.
Comparison of the economic effects of the technologies under consideration is not provided.
As per the reviewer comments, a new section has been added.
There is no comparison of the productivity of the technologies under consideration. A detailed comparison of productivity (l / m2 / day or kg / m2 / day) should be given in the Results and Discussion section.
As per the reviewer comments, results and discussion has been enhanced..
There is a very lack of specifics (an example of the application of technology with an indication of the country and city).
As per the reviewer comments, results and discussion has been enhanced..
Table 1 contains a comparison of all considered methods? If so, why is it listed in the “Desiccant Systems” section? I recommend that you pay attention to its detailed consideration in the "Results and Discussion" section.
As per the reviewer comments, correction has been made.
I recommend adding a section "Acknowledgments".
The "Conclusions" section should be expanded. I recommend adding to it a comparison of specific indicators of all the technologies considered.
As per the reviewer comments, conclusion has been strengthened.
The authors would like to thank the reviewers and editor for specifically pointing out that the flaws in the research article which is much needed information.
Thank you very much for your valuable suggestions
Reviewer 2 Report
The topic is interesting from the point of view of topic concentrated on water extraction from the atmospheric air. There are lack of articles on this topic, therefore information with reference to presented examples on this subject is valuable.
The study has good potential and the manuscript is well written.
Despite this, I have 2 comments regarding redrafting of the manuscript:
- The introduction lacks information about the novelity and the global significance of the presented research results. I mean, how other authors could benefit from the solutions presented by you.
- It is necessary to refer to a broader approach to the subject by comparing it to similar research from a global perspective. It is necessary to cite current articles of world famous authors who research this topic.
Author Response
The authors would like to place their sincere thankfulness to the Editor, Reviewers and Coordinators of the Energies for considering our paper for its publication and further providing their valuable suggestions.
The answers for the queries raised by the reviewers are answered herewith clarity and humbleness.
Comments to the Author
Reviewer 2:
Comments and Suggestions for Authors
The topic is interesting from the point of view of topic concentrated on water extraction from the atmospheric air. There are lack of articles on this topic, therefore information with reference to presented examples on this subject is valuable.
The study has good potential and the manuscript is well written.
Despite this, I have 2 comments regarding redrafting of the manuscript:
The introduction lacks information about the novelity and the global significance of the presented research results. I mean, how other authors could benefit from the solutions presented by you.
As per the reviewer comments, it has enhanced.
It is necessary to refer to a broader approach to the subject by comparing it to similar research from a global perspective. It is necessary to cite current articles of world famous authors who research this topic.
As per the reviewer comments, recent articles have been added.
The authors would like to thank the reviewers and editor for specifically pointing out that the flaws in the research article which is much needed information.
Thank you very much for your valuable suggestions
Reviewer 3 Report
This paper provides a comprehensive review on the extraction of water from atmosphere in arid zones. This paper is valuable and well written and organized.
This paper could be made more valuable by adding a section with name “insights for “policymakers and practitioners”. Please add a section before conclusion. This section should include detailed opinions and recommendations written by authors based on strong evidence.
Author Response
The authors would like to place their sincere thankfulness to the Editor, Reviewers and Coordinators of the Energies for considering our paper for its publication and further providing their valuable suggestions.
The answers for the queries raised by the reviewers are answered herewith clarity and humbleness.
Comments to the Author
Reviewer 3:
Comments and Suggestions for Authors
This paper provides a comprehensive review on the extraction of water from atmosphere in arid zones. This paper is valuable and well written and organized.
This paper could be made more valuable by adding a section with name “insights for “policymakers and practitioners”. Please add a section before conclusion. This section should include detailed opinions and recommendations written by authors based on strong evidence.
As per the reviewer comments, correction have been done.
The authors would like to thank the reviewers and editor for specifically pointing out that the flaws in the research article which is much needed information.
Thank you very much for your valuable suggestions
Reviewer 4 Report
Topic
The topic is interesting.
Introduction
Highlight the need for the study and the importance of the study in the face of existing studies.
Literature Review
Figure 1 is dispensable.
In the literature review the reader does not understand what method was followed.
Methodology
There is no methodology section that allows the reader to understand the study and replicate it.
Conclusions
The conclusions are in line with the study.
Highlight the limitations of the study and suggestions for future research.
Author Response
The authors would like to place their sincere thankfulness to the Editor, Reviewers and Coordinators of the Energies for considering our paper for its publication and further providing their valuable suggestions.
The answers for the queries raised by the reviewers are answered herewith clarity and humbleness.
Comments to the Author
Reviewer 4:
Comments and Suggestions for Authors
Topic
The topic is interesting.
Introduction
Highlight the need for the study and the importance of the study in the face of existing studies.
As per the reviewer comments, correction have been done.
Literature Review
Figure 1 is dispensable.
As per the reviewer comments, it has been removed and replaced with an opted figure..
In the literature review the reader does not understand what method was followed.
The literature review was strengthened.
Methodology
There is no methodology section that allows the reader to understand the study and replicate it.
As per the reviewer comments, correction have been done.
Conclusions
The conclusions are in line with the study.
Highlight the limitations of the study and suggestions for future research.
As per the reviewer comments, they have been included
The authors would like to thank the reviewers and editor for specifically pointing out that the flaws in the research article which is much needed information.
Thank you very much for your valuable suggestions
Round 2
Reviewer 4 Report
When a revised version is presented, generally, to aid the reviewer's reading, changes are highlighted in a different color.
This encourages comparison between the two versions presented: the previous and the revised version.
However, this is not the case in this version.
In addition, the changes made do not meet the recommendations made.
Author Response
The authors would like to place their sincere thankfulness to the Editor, Reviewers and Coordinators of the Energies for considering our paper for its publication and further providing their valuable suggestions.
The answers for the queries raised by the reviewers are answered herewith clarity and humbleness.
Comments to the Author
Reviewer 4:
When a revised version is presented, generally, to aid the reviewer's reading, changes are highlighted in a different color.
This encourages comparison between the two versions presented: the previous and the revised version.
However, this is not the case in this version.
In addition, the changes made do not meet the recommendations made.
- As per the reviewer comments, changes are highlighted in a different color.
Suggestion for future research have been added and the conclusion has been strengthened.
The authors would like to thank the reviewers and editor for specifically pointing out that the flaws in the research article which is much needed information.
Thank you very much for your valuable suggestions
Round 3
Reviewer 4 Report
It is not possible to approve a scientific paper without a methodology section.
Highlight the limitations of the study and suggestions for future research.
Author Response
The authors would like to place their sincere thankfulness to the Editor, Reviewers and Coordinators of the Energies for considering our paper for its publication and further providing their valuable suggestions.
The answers for the queries raised by the reviewers are answered herewith clarity and humbleness.
Comments to the Author
Reviewer 1:
The answers for the queries raised by the reviewers are answered herewith clarity and humbleness. Authors thank the reviewer for the extended care to enhance the manuscript to a good quality article,
It is not possible to approve a scientific paper without a methodology section.
-As per the reviewer suggestion, it is kindly said that methodology is not included in the current manuscript since it is fully written as review article.
Highlight the limitations of the study and suggestions for future research.
-As per the reviewer suggestion, a new section namely future research has been included in the manuscript.
Thank you very much for your valuable suggestions